# Comparative Transcriptome Analyses of Leg Muscle during Early Growth between Geese (*Anser cygnoides*) Breeds Differing in Body Size Characteristics

**DOI:** 10.3390/genes14051048

**Published:** 2023-05-07

**Authors:** Jun Tang, Hongjia Ouyang, Xiaomei Chen, Danli Jiang, Yunbo Tian, Yunmao Huang, Xu Shen

**Affiliations:** 1College of Animal Science and Technology, Zhongkai University of Agriculture and Engineering, Guangzhou 510225, China; tangjun841918@126.com (J.T.); oyhj@zhku.edu.cn (H.O.); 13413663307@163.com (X.C.); danli0222@163.com (D.J.); tyunbo@126.com (Y.T.); huangyunmao@163.com (Y.H.); 2Waterfowl Healthy Breeding Engineering Research Center, Guangdong Higher Education Institutes, Guangzhou 510225, China

**Keywords:** goose, growth performance, growth curve, transcriptome

## Abstract

Goose is an important poultry commonly raised for meat. The early growth performance of geese significantly influences their market weight and slaughter weight, affecting the poultry industry’s economic benefits. To identify the growth surge between the Shitou goose and the Wuzong goose, we collected the early growth body traits from 0 to 12 weeks. In addition, we investigated the transcriptomic changes in leg muscles at the high growth speed period to reveal the difference between the two geese breeds. We also estimated the growth curve parameters under three models, including the logistic, von Bertalanffy, and Gompertz models. The results showed that except for body length and keel length, the best-fitting model between the body weight and body size of the Shitou and Wuzong was the logistic model. The growth turning points of Shitou and Wuzong were 5.954 and 4.944 weeks, respectively, and the turning point of their body weight was 1459.01 g and 478.54 g, respectively. Growth surge occurred at 2–9 weeks in Shitou goose and at 1–7 weeks in Wuzong goose. The body size traits of the Shitou goose and Wuzong goose showed a trend of rapid growth in the early stage and slow growth in the later stage, and the Shitou goose growth was higher than the Wuzong goose. For transcriptome sequencing, a total of 87 differentially expressed genes (DEGs) were identified with a fold change ≥ 2 and a false discovery rate < 0.05. Many DEGs have a potential function for growth, such as *CXCL12*, *SSTR4*, *FABP5*, *SLC2A1*, *MYLK4*, and *EIF4E3*. KEGG pathway analysis identified that some DEGs were significantly enriched in the calcium signaling pathway, which may promote muscle growth. The gene–gene interaction network of DEGs was mainly related to the transmission of cell signals and substances, hematological system development, and functions. This study can provide theoretical guidance for the production and breeding management of the Shitou goose and Wuzong goose and help reveal the genetic mechanisms underlying diverse body sizes between two goose breeds.

## 1. Introduction

China is the largest goose producer and market for meat geese in the world. In 2021, 570 million commercial geese were sold, and the output value of meat geese is 52.26 billion yuan, 6.8% higher than that in 2020 [1]. The production of meat for poultry is closely related to growth performance, which is an important economic trait.

Geese have the fastest initial growth rate of poultry [2], and the growth rate of geese varies with the growth period. As a nonlinear curve model, a growth curve is an effective way to depict the patterns of growth and development in animals [3,4]. The model can compress data, such as age, weight, and body size, into several parameters to eliminate the influence of experimental errors. At present, three nonlinear growth models, namely, logistic, Gompertz, and von Bertalanffy, are usually used to research the growth and development patterns of birds [5,6,7]. The early growth and development of geese have important effects on the time to market and slaughter weight. At present, studies on the early growth of poultry are mainly based on the chicken model, but few are based on geese.

Growth traits are critical economic traits in the poultry industry that are controlled by multiple genes and the environment [8]. In recent years, RNA sequencing has been widely applied to reveal the molecular mechanisms of growth performance in chickens, but little RNA sequencing has been applied to geese. RNA-seq technology provides the opportunity to assess global changes in the transcriptome and to understand regulatory pathways during the early growth of geese [9,10].

The Shitou goose is the largest goose species in the world, with many advantages, such as fast growth, enormous weight, and affluent muscle [2]. The average weight of the male goose can reach 10.39 kg [11], and its growth performance is excellent. Wuzong goose is the smallest goose species, and the male goose weighs only 3–3.5 kg [12]. Their growth performance is very different, which is an ideal model for studying the growth and development of local geese in China. Because the goose has no keel protrusion, the leg muscle rate of the goose is higher than the chest muscle rate; therefore, the regulatory mechanisms for the growth and development of goose leg muscle is very important.

In this study, the body weight and body size traits of 0–12 week-old Shitou and Wuzong geese were measured, and the growth curves of the body weight and body size traits were fitted and analyzed using the logistic, von Bertalanffy, and Gompertz models. In order to explore the genetic mechanisms underlying goose growth performance, we applied comparative transcriptome analysis of large-size Shitou goose with small-size Wuzong goose at the age of peak growth. The results of this study are useful for understanding the mechanisms regulating the development of leg muscle and the pattern of goose growth.

## 2. Materials and Methods

### 2.1. Ethics Statement

All experimental animal procedures complied with the laboratory animal management and welfare regulations approved by the Experimental Animal Committee of Zhongkai University of Agriculture and Engineering, Guangzhou, Guangdong, China. This study was approved by the Experimental Animal Committee of Zhongkai University of Agriculture and Engineering (NO.2019110220). All efforts were made to minimize animal suffering.

### 2.2. Animals and Sample Preparation

The largest-size Shitou goose and the smallest-size Wuzong goose from Qingyuan Jinyufeng Goose Industry Co., Ltd. (Qingyuan, China) were used in this experiment. Forty one-day-old Shitou geese and forty one-day-old Wuzong geese were randomly selected and fed to twelve weeks of age. All animals used in this experiment were raised in free-range under the same conditions with a free diet and supplemented with green grass. Three male Shitou geese and three male Wuzong geese were selected and slaughtered at 5 weeks of age. After slaughtering, leg muscle was collected for transcriptome sequencing and real-time PCR validation.

### 2.3. Measurement Index and Method

The measurement of body weight and body size traits refers to the statistical method for measuring poultry production performance issued by the Ministry of Agriculture of China (NY/T 823-2004, Beijing, China). The measurement items include body weight, body length, chest depth, chest width, pelvis width, keel length, tibia length, shank circumference, semi-submersible length (waterfowl), and neck length.

### 2.4. Statistical Analysis

The SPSS 17.0 software package (SPSS Inc., Chicago, IL, USA) was used to analyze the data. Three nonlinear growth models, including logistic, Gompertz, and von Bertalanffy, were used to fit the growth curves of the geese [5,6,7]. The equations of these three nonlinear models are shown in Table 1. GraphPad Prism 5.0 software (GraphPad Software Inc., San Diego, CA, USA) was used to draw the growth curve.

### 2.5. Library Preparation and Transcriptome Sequencing

According to the manufacturer’s instructions, total RNA was extracted from the leg muscle tissues with Trizol reagent (Invitrogen, Carlsbad, CA, USA). RNA purity was checked with the RNeasy Animal Mini Kit (Qiagen, Valencia, CA, USA), and RNA quality and concentration were checked with Agilent 2100 BioAnalyzer (Agilent, Santa Clara, CA, USA) and QUBIT RNA ASSAYKIT (Invitrogen). Subsequently, genomic DNA was removed by using RNase-Free DNase I (Qiagen). The total RNA of each individual was used to construct a cDNA library for RNA-seq. Six cDNA libraries were prepared and were sequenced through the Illumina HiSeq2000 system (Novogene, Beijing, China).

### 2.6. Bioinformatic Analysis of the RNA-Seq Data

Raw reads were filtered by removing adaptors and low-quality reads with Trim Galore. Subsequently, the clean reads were mapped to the reference genome (Anser cygnoides domesticus, AnsCyg_ PRJNA183603_v1.0) [13] using TopHat2 software with the default parameters [14]. FPKM (fragments per kilobase per million reads) was used to quantify the expression level. The DESeq2 package was used to calculate the differences in gene expression by the criteria that fold change ≥ 2 and a false discovery rate (FDR) ≤ 0.05. The Gene Ontology enrichment of DEGs was analyzed with GOseq software [15]. Kyoto Encyclopedia of Genes and Genomes (KEGG) pathway analysis of the DEGs was performed with the software DAVID [16]. The gene–gene interaction network [17] of DEGs was built up using the ingenuity pathway analysis (IPA) database (http://www.ingenuity.com, accessed on 1 September 2019).

### 2.7. Quantitative Reverse Transcription Real-Time PCR Reverse Transcription (qRT-PCR) Validation

To verify the reliability of the RNA sequencing data, 11 DEGs were randomly selected for qRT-PCR validation. The primers were designed using Primer Premier 5.0 software (Appendix A). qRT-PCR was performed in three repetitions for each sample. The relative expression levels of DEGs were calculated using the 2^−ΔΔCt^ method with *GAPDH* as the control [18]. The reaction conditions were as follows: 95 °C for 10 min, 40 cycles of 95 °C for 15 s, and 60 °C for 60 s.

## 3. Results

### 3.1. Fitting and Analysis of Three Nonlinear Growth Models for the Body Weight and Body Size Traits of the Shitou Goose and Wuzong Goose

The logistic, von Bertalanffy, and Gompertz models can fit the growth curves of the body weight and body size traits of the Shitou goose and the Wuzong goose well, with an R^2^ higher than 0.96 (Appendix A). The best fitting model of the body weight and body size of the Shitou goose and Wuzong goose is the logistic model, except for the body length and keel length of Wuzong goose, for which the best model is Bertalanffy (Table 2). The weeks of age and value of body weight and body traits at the inflection point of growth in the best-fitted model are shown in Table 2. The fitting results closely matched the measured values.

### 3.2. Body Weight Growth Curve and Relative Growth Rate of Shitou Goose and Wuzong Goose

The results showed that the body weight growth curve of the Shitou goose and Wuzong goose showed an S shape (Figure 1). The Shitou goose gained weight rapidly from the second week, with a relative growth rate of 113.42%. The age of the inflection point was 5.954 weeks (Table 2). The growth rate reached the fastest around the age of 5 weeks, with a maximum weekly weight gain of 789.97 g. The growth rate from 2 to 9 weeks was very fast, with a weekly weight gain of more than 220 g and a relative growth rate of more than 11% (Table 3). The relative growth rate gradually decreased with the increase in weeks of age, and the growth rate slowed down rapidly after the ninth week. The Wuzong goose gained weight rapidly from the first week, with a relative growth rate of 165.15% and an inflection point age of 4.944 weeks (Table 2). The growth rate reached the fastest around the age of 5 weeks, with a maximum weekly weight gain of 478.54 g. The growth rate from 1 to 7 weeks was very fast, with weekly weight gains of more than 107 g and a relative growth rate of more than 20%.

### 3.3. Growth Curve of Body Size Traits of Shitou Goose and Wuzong Goose

The body size traits of the Shitou goose and Wuzong goose showed a trend of rapid growth in the early stage and slow growth in the later stage, and the Shitou goose growth rate was higher than the Wuzong goose (Figure 2). The body length, semi-submersible length, and neck length of the Shitou goose increased rapidly before the age of 7 weeks and gradually slowed down after the age of 7 weeks. The body length and semi- submersible length grew very fast from 3 to 6 weeks, and the neck length grew very fast from 3 to 5 weeks. The growth curve of the shank circumference, tibia length, and pelvis width presented a typical para-curve. The shank circumference and tibia length grew rapidly in the first 5 weeks, and the pelvis width grew rapidly in the first 5 weeks, and then the growth was very slow. The chest depth, chest width, and keel length grew rapidly and steadily in the first 7 weeks.

The rule of growth and development of the Wuzong goose was similar to that of the Shitou goose, and the rapid growth period of its traits was shorter than that of Shitou goose. The body length and semi-submersible length increased rapidly before 5 weeks, and the neck length increased rapidly before 6 weeks. The tibia length and shank circumference grew very fast from 0 to 4 weeks, and the pelvis width grew very fast from 0 to 5 weeks. The chest depth, chest width, and keel length grew faster in the first five weeks and slowed down slightly after five weeks.

### 3.4. The Body Weight of Shitou Goose and Wuzhong Goose Used in RNA-Seq

The inflection point age of the body weight of the Shitou goose was 5.954 weeks, and the inflection point age of the body weight of the Wuzhong goose was 4.944 weeks, and the fastest growing period of both geese was around 5 weeks. Therefore, in this study, the 5-week-old Shitou goose and Wuzhong goose were selected to study. At this time, the body weight difference between the two geese is extremely significant. See the previous study [19] for details.

### 3.5. The Transcriptome Profile of the Leg Muscle between Two Breeds

In this study, a total of 6 cDNA libraries were constructed from the leg muscle of the Shitou goose (TS1, TS2, TS3) and the Wuzong goose (TW1, TW2, TW3). The clean data from each sample reached more than 6.66 Gb. The percentage of Q30 base in each sample was greater than 92.36%, and the GC content reached 50.87–52.02% (Table 4). The comparison efficiency of the total reads compared to the reference genome of the six samples was between 57.02 and 59.24%, and the percentage of reads compared to the only location of the reference genome was between 56.35 and 58.52% (Table 4) in the clean reads. The above results indicated that the data were reliable to be used for further analysis.

### 3.6. Identification of Differentially Expressed Genes

A total of 87 DEGs (Appendix A) were identified by fold change ≥ 2 and FDR < 0.05, and there were 39 upregulated genes and 48 downregulated genes in the Shitou goose compared to the Wuzong goose. The difference in gene expression between the two groups and its statistical significance can be seen from the volcanic map (Figure 3).

### 3.7. GO and KEGG Pathway Analyses of DEGs

GO enrichment analysis was used to annotate the function of DEGs. The GO enrichment includes three categories: biological process, cellular component, and molecular function (Figure 4). In terms of molecular function, the DEGs were mainly related to transferase activity, transferring glycosyl groups, and transferring hexosyl groups. Four DEGs, including *ST8SIA6*, *B3GAT2*, *SSTR4*, and *LOC106030438*, were enriched for GO terms involved in the growth and development of muscle.

Pathway enrichment analysis was performed to gain insight into the functional roles of DEGs in the leg muscle between the two goose species. Four significantly enriched pathways were identified, including the metabolism of xenobiotics by cytochrome P450, glutathione metabolism, the calcium signaling pathway, and vascular smooth muscle contraction. There were seven DEGs involved in the four pathways (Table 5).

### 3.8. Gene–Gene Interaction Network

The gene–gene interaction network between DEGs was analyzed with IPA software. A total of 12 interaction networks (Appendix A) were formed from 87 DEGs. The most significant enriched network was related to cell-to-cell signaling and interaction, cellular assembly and organization, and hematological system development and function, which consisted of 6 upregulated genes, including *ADORA2A*, *COL4A5*, *CXCL12*, *FABP5*, *PCDH19*, and *SLC2A1*, and 10 downregulated genes, including *ANKRD1*, *EZR*, *HSPB1*, *MAP3K4*, *NOP16*, *ODC1*, *P38 MAPK*, *RPS15*, *RSAD2*, and *THBD* (Figure 5).

### 3.9. Validation of DEGs by qPCR

To verify the reliability of the sequencing data, 11 DEGs were randomly selected for qPCR verification. The quantitative results of qPCR were highly correlated with the RNA-seq data (Figure 6), confirming the accuracy of the results of the RNA-seq.

## 4. Discussion

Growth traits are one of the most important economic traits in the commercial poultry industry. Body weight and body size are important indicators of poultry growth traits. The rule of growth and the development of body weight and body size is suitable for fitting with the nonlinear model. At present, the widely used nonlinear growth curve models are mainly the logistic, von Bertalanffy, and Gompertz models [20,21]. Different varieties have different best-fitting nonlinear models because of different growth and development rules. In this study, the best fitting model for the body weight of the Shitou goose and Wuzong goose was the logistic model, with R^2^ 0.999 and 0.998, respectively, and the fitting result was the closest to the measured value. The growth inflection point is the time point of the fastest growth of animals and the turning point of the growth rate from fast to slow. The earlier inflection point time indicates faster early growth speed in the animals and earlier body and sexual maturity times, which can shorten the time to market. In the present study, our results showed that the inflexion ages of the body weight of the Shitou goose and the Wuzong goose were estimated at 5.954 and 4.944 weeks of age, respectively, and the absolute body weight gain in the two breeds were 789.97 g and 478.54 g, respectively. Five weeks of age was the fastest growth period in the two breeds, of which the weekly weight gain reached the maximum. In addition, the weekly weight gain of the Shitou goose at the age of 2 to 9 weeks was more than 220 g, and the relative growth rate was more than 11%. The weekly weight gain of the Wuzong goose at the age of 1 to 7 weeks was more than 107 g, and the relative growth rate was more than 20%, which indicates that the Shitou goose and the Wuzong goose grow rapidly at the early stage. At this time, the Shitou goose and the Wuzong goose should be strengthened in nutrition and scientific management to give full support to their growth potential, improve their market weight, and shorten the time to market to improve the economic benefits of farmers.

Body size is also an important indicator to measure the growth and development of animals. In this study, the body length and semi-submersible length of the Shitou goose had the fastest growth rate between 3 and 6 weeks, and the neck length had the fastest growth rate between 3 and 5 weeks. The growth of the longitudinal body size lays the foundation for weight gain. Tibia length and shank circumference had the fastest growth rate in the first 5 weeks. A better growing tibia can better support weight and exercise. The chest deep, chest width, keel length, and pelvis width showed very rapid growth in the first 7 weeks. The rapid growth period of each index of the Wuzong goose was shorter than that of the Shitou goose. The body length and semi-submersible length grew rapidly before 5 weeks, and the neck length grew rapidly before 6 weeks. The rapid growth period of the tibia length and shank circumference was between 0 and 4 weeks, and the pelvic width was between 0 and 5 weeks. The chest depth, chest width, and keel length grew faster in the first five weeks. Therefore, Shitou goose in the first 7 weeks and Wuzong goose in the first 5 weeks should have strengthened supplements of various nutrients in their feed, especially the content of protein, calcium, phosphorus, and other mineral elements, in order to promote bone growth and prevent the occurrence of leg disease.

The growth and development of poultry was strongly controlled with a combination of genetics, environment factors, and nutrition [8]. The identification of candidate genes related to poultry growth traits has been the topic of extensive investigation. In China, except for Yili geese, which originated from the greylag goose (Anser anser), all the local geese originate from the swan goose (Anser cygnoides) [22]. The Shitou goose and the Wuzong goose are genetically similar in that they originated from the same ancestor and lived in the same habitat [23]. However, the two breeds showed a marked difference in body size. The body weight of an adult Wuzong goose was less than one-third of an adult Shitou goose, and a significant difference in growth performance was observed between the two breeds. As growth involves the development of muscles, people study the differences in growth rates in birds primarily through muscle development [24,25].

In this study, transcriptomes of the leg muscles were compared between the Shitou goose and the Wuzong goose at 5 weeks, which is the growth peak period for both Shitou goose and Wuzong goose to identify potential genes and pathways causing differences in growth and development. Multiple mapping reads were slightly lower in our data, and we focused on unique mapping reads to obtain reliable gene expression results. We identified 87 DEGs, including 39 upregulated and 48 downregulated. Some of these DEGs, such as *CXCL12*, *SSTR4*, *FABP5*, *SLC2A1*, *MYLK4*, and *EIF4E3*, were related to growth.

Somatostatin (SST), as a signal molecule, is mediated by the family of SST receptors (SSTRs) on the cell membrane and plays an important role in inducing cell apoptosis, inhibiting tumor cell proliferation, inhibiting the function of insulin, and inhibiting cell growth and other biological processes [26,27,28], such as SSTR-mediated inhibition of secretion and cell proliferation. Therefore, SSTR4 can inhibit the growth hormone and insulin release, hindering cell growth and expansion. In this study, the expression level of *SSTR4* in the Wuzong goose was significantly higher than that in the Shitou goose, which may have an inhibiting effect on the development of the Wuzong goose.

*CXCL12* is crucial for promoting developmental myogenesis and plays an essential role in muscle growth and angiogenesis in skeletal muscle [29,30]. *CXCL12* was recognized to play a major role in the maintenance, development, and differentiation of progenitor stem cells in the musculoskeletal system [31]. In this study, the expression of *CXCL12* in the Shitou goose was higher than that in the Wuzong goose, which may promote muscle development and growth in the Shitou goose.

Eukaryotic translation initiation factor 4E (*eIF4E*), which is considered the cornerstone in the cap-dependent translation initiation machinery, is implicated in cell transformation, tumorigenesis, and angiogenesis by facilitating the translation of oncogenic mRNAs [32]. However, *eIF4E3*, whose function is dependent on its atypical cap-binding activity, is an inhibitor rather than a promoter of both target transcript expression and oncogenic transformation [33]. The expression of *eIF4E3* in the Wuzong goose was higher than that in the Shitou goose, which may inhibit growth in the Wuzong goose.

GO analysis showed that the DEGs were enriched in glutathione γ-glutamyl cysteinyl transferase activity. Additionally, four significant pathways, including the metabolism of xenobiotics by cytochrome P450, glutathione metabolism, calcium signaling pathway, and vascular smooth muscle contraction, were identified by KEGG analysis.

Calcium is a second messenger critical for many cellular processes including proliferation which is a complex mechanism orchestrated by several proteins related to Ca^2+^ signaling [34,35], migration, and vesicular transport [36]. High external Ca^2+^ concentrations can trigger cell proliferation in normal cells [37,38].

Ca^2+^ signaling is essential during mitotic and meiotic cell cycles to break down the nuclear envelope to promote cell division [39,40]. Ca^2+^ signaling also regulates various aspects of cell cycle transition and cellular proliferation during the cell cycle through downstream Ca^2+^-dependent signaling modules [41,42]. At the same time, proliferation induced by store-independent Ca^2+^ signaling seems to be an important mechanism of tumorigenesis [43,44]. In this study, three genes related to the calcium signaling pathway upregulated in the Shitou goose, which may promote muscle growth.

Three critical gene networks were identified through the analysis of the gene interaction network. The first network is related to cell-to-cell signaling and interaction, cellular assembly and organization, and hematological system development and function, in which *COL4A5*, *CXCL12*, *FABP5*, and *SLC2A1* were up-regulated in the Shitou goose. Glucose transporter protein type 1 (GLUT-1) encoded by the *SLC2A1* gene mediate the facilitated transport of glucose, which is the primary energy substance of tissue cells, and the striated skeletal muscle in birds is a central glucose-utilizing tissue. Therefore, the *SLC2A1* gene can induce skeletal regeneration [45]; at the same time, GLUT-1 deficiency causes developmental delay [46]. GLUT-1 is the primary expression of red blood cells and is responsible for glucose transport between various tissues and blood. Its main function is to maintain the glucose intake of tissues and cells in the basic state, which plays a crucial role in maintaining the stability of blood glucose concentration. It is essential for the development and function of the blood system. A sound hematological system can provide a large number of nutrients for the growth of body tissues and transport metabolic waste.

The function of FABP is to store and transport fatty acids in cells for a short time and participate in the transport and metabolism of fatty acids. It can transport fatty acids from the cell membrane to the sites of fatty acid oxidation, triglyceride, and phospholipid synthesis [47]. *FABP5* is the delivery of ligands to *PPARd*, which promotes cell migration, proliferation, and survival [48,49,50]. Therefore, *FABP5* can promote cell growth and proliferation through the transmission of cell signals and substances, which also promotes the growth and development of the body.

## 5. Conclusions

In this study, we determined the time of rapid growth between Shitou geese and the Wuzhong geese. We identified 87 DEGs by RNA-seq, some which have a potential function on growth, such as *CXCL12*, *SSTR4*, *FABP5*, *SLC2A1*, *MYLK4*, and *EIF4E3*. KEGG pathway analysis identified that some DEGs significantly enriched in the calcium signaling pathway, which may promote muscle growth. The interaction network of DEGs was mainly related to the transmission of cell signals and substances, hematological system development, and function. The results of this study will help understand the genetic mechanisms of goose growth and development.

## Figures and Tables

**Figure 1 genes-14-01048-f001:**
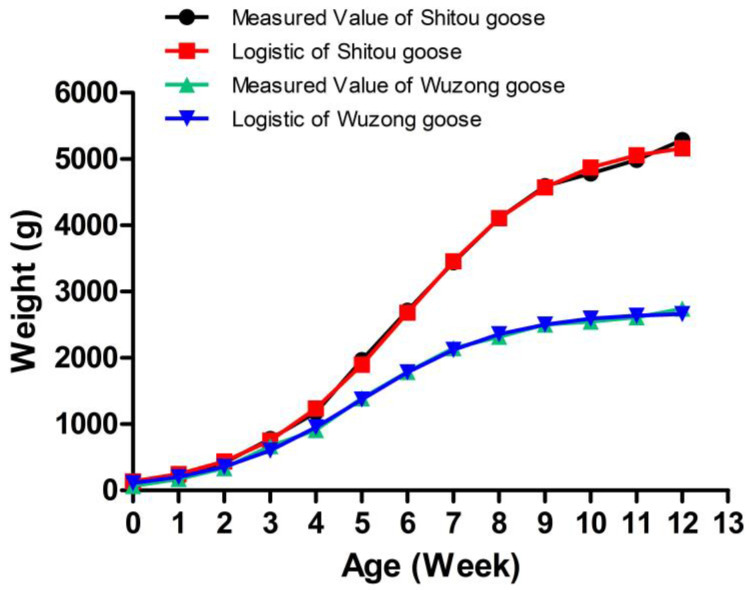
Growth curve of measured body weight and the best fitting model of the Shitou goose and Wuzong goose.

**Figure 2 genes-14-01048-f002:**
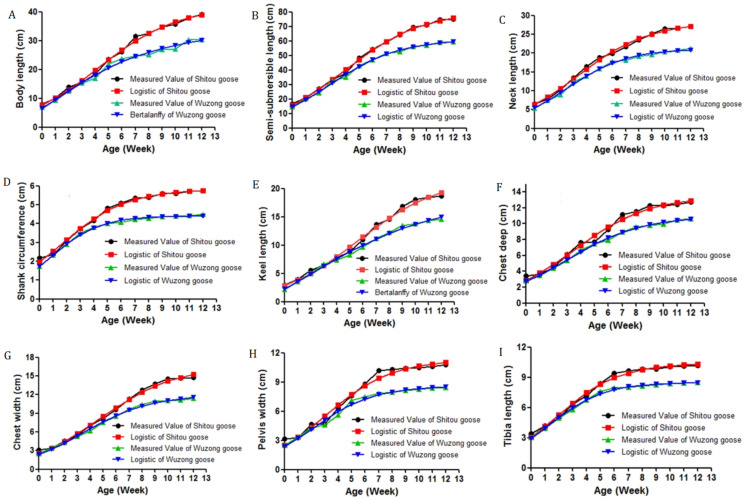
The actual growth curve and the matching curve of the Shitou and Wuzong goose. (**A**–**I**) represent growth curve of the Body length, Semi-submersible length, Neck length, Shank circumference, Keel length, Chest deep, Chest width, Pelvis width, Tibia length, respectively.

**Figure 3 genes-14-01048-f003:**
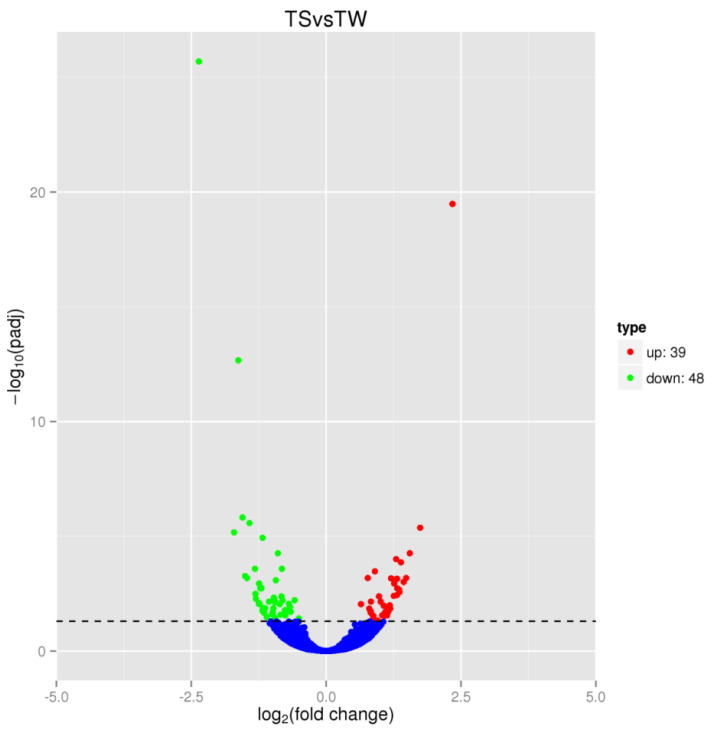
Volcano plot displaying differentially expressed genes. The *y*-axis corresponds to the mean of the expression value of log10 (*q*-value), and the *x*-axis presents the log2 fold change value. The red dot represents up-regulated transcripts, and the green dots indicate down-regulated transcripts (*q* < 0.05), while the blue dots mean the transcripts failed to reach statistical significance (*q* > 0.05).

**Figure 4 genes-14-01048-f004:**
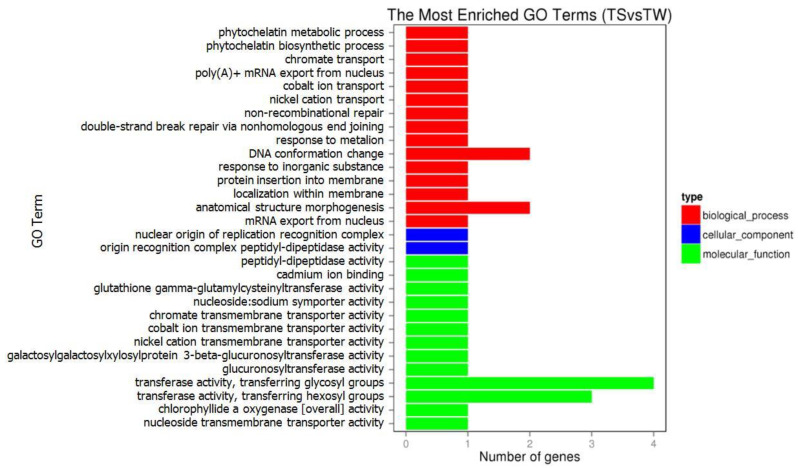
GO enrichment analysis of DEGs identified in the leg muscle between the two goose species.

**Figure 5 genes-14-01048-f005:**
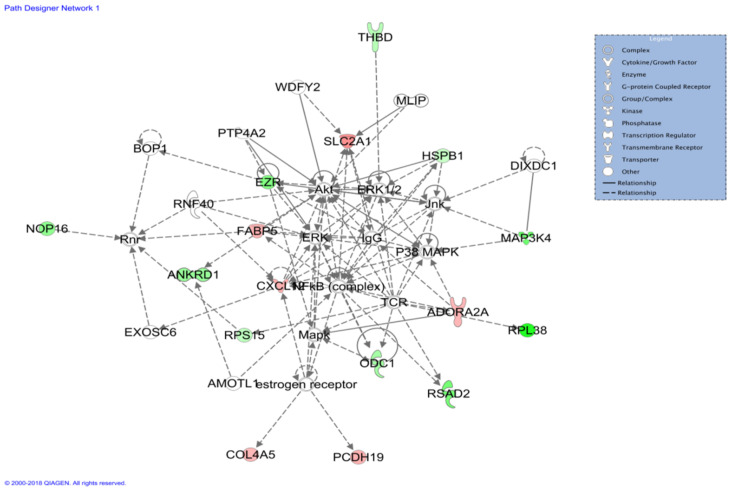
IPA network related to cell-to-cell signaling and interaction, cellular assembly and organization, and hematological system development and function. Genes colored in red represent up-regulated in TS geese, while genes colored in green are down-regulated in TS geese. The color intensity is proportional to the fold change of DEGs.

**Figure 6 genes-14-01048-f006:**
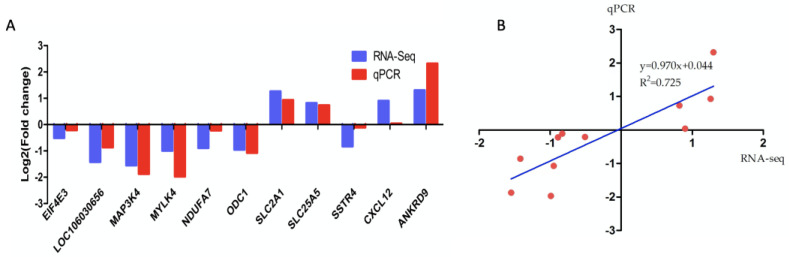
qPCR validation of DEGs obtained from the RNA-seq data in TS–VS–TW. (**A**) Comparison expression differences were obtained from RNA-seq data (blue) and qPCR data (red). (**B**) Lineage analysis between the RNA-seq and RT-qPCR data.

**Table 1 genes-14-01048-t001:** Three nonlinear growth models.

Model Name	Model	Inflection Point Value	Inflexion Age	Maximum Age Increment
Logistic	Y = A/(1 + Be^−kt^)	A/2	(lnB)/k	kw/2
Von Bertalanffy	Y = A(1 − Be^−kt^)^3^	8 A/27	(ln3B)/k	3 kw/2
Gompertz	Y = Ae ^−Bexp(−kt)^	A/e	(lnB)/k	kw

Note: A is the upper asymptote (growth limit) parameter, k is the instantaneous growth rate, B is the adjustment parameter, t is the age in weeks, and w is the inflection point value.

**Table 2 genes-14-01048-t002:** Results of the estimates for the best-fitting nonlinear models.

Items	Goose	Model	A	B	K	R^2^ *	WGI ^a^	BGI ^b^
Body weight, g	Shitou goose	Logistic	5290.870	37.334	0.608	0.999	5.954	2645.435
	Wuzong goose	Logistic	2692.975	23.443	0.638	0.998	4.944	1346.488
Body length, cm	Shitou goose	Logistic	41.720	4.421	0.344	0.995	4.321	20.860
	Wuzong goose	Bertalanffy	33.649	0.420	0.204	0.989	1.133	9.970
Chest depth, cm	Shitou goose	Logistic	13.486	3.724	0.369	0.987	3.563	6.743
	Wuzong goose	Logistic	10.982	3.041	0.365	0.998	3.047	5.491
Chest width, cm	Shitou goose	Logistic	16.465	5.411	0.350	0.995	4.824	8.233
	Wuzong goose	Logistic	11.972	3.966	0.388	0.996	3.551	5.986
Pelvis width, cm	Shitou goose	Logistic	11.298	3.569	0.409	0.980	3.111	5.649
	Wuzong goose	Logistic	8.580	2.603	0.445	0.987	2.150	4.290
Keel length, cm	Shitou goose	Logistic	21.676	6.438	0.329	0.993	5.660	10.838
	Wuzong goose	Bertalanffy	18.558	0.501	0.165	0.995	2.469	5.499
Tibia length, cm	Shitou goose	Logistic	10.385	2.387	0.451	0.992	1.929	5.193
	Wuzong goose	Logistic	8.499	1.891	0.504	0.997	1.264	4.250
Shank circumference, cm	Shitou goose	Logistic	5.835	1.982	0.420	0.994	1.629	2.918
	Wuzong goose	Logistic	4.405	1.529	0.554	0.996	0.766	2.203
Semi-submersible length, cm	Shitou goose	Logistic	80.536	4.030	0.348	0.998	4.005	40.268
	Wuzong goose	Logistic	61.177	3.128	0.391	0.997	2.917	30.589
Neck length, cm	Shitou goose	Logistic	28.192	3.455	0.369	0.996	3.360	14.096
	Wuzong goose	Logistic	21.191	2.891	0.429	0.996	2.475	10.596

* R2 represents the goodness-of-fit. ^a^ WGI indicates the weeks of age at the inflexion point of growth. ^b^ BGI indicates the value at the inflexion point of growth.

**Table 3 genes-14-01048-t003:** Body weight growth and relative growth rates of Shitou goose and Wuzong goose.

Geese	Items	0 Weeks	1 Week	2 Weeks	3 Weeks	4 Weeks	5 Weeks	6 Weeks	7 Weeks	8 Weeks	9 Weeks	10 Weeks	11 Weeks	12 Weeks
Shitou geese	Body weight	123.55	194.60	415.32	775.65	1176.13	1966.10	2710.08	3444.15	4109.31	4594.38	4780.85	4989.92	5289.92
	Weight gain per week		71.05	220.72	360.33	400.47	789.97	743.98	734.08	665.15	485.08	186.46	209.08	300.00
	Relative growth rate		57.50%	113.42%	86.76%	51.63%	67.17%	37.84%	27.09%	19.31%	11.80%	4.06%	4.37%	6.01%
Wuzong geese	Body weight	65.39	173.37	337.80	669.63	911.83	1390.37	1786.33	2146.17	2317.60	2503.88	2545.63	2612.32	2743.04
	Weight gain per week		107.98	164.43	331.83	242.20	478.54	395.96	359.84	171.43	186.28	41.74	66.70	130.72
	Relative growth rate		165.15%	94.84%	98.23%	36.17%	52.48%	28.48%	20.14%	7.99%	8.04%	1.67%	2.62%	5.00%

**Table 4 genes-14-01048-t004:** Summary of the transcripts in goose muscle tissues.

Sample	Raw Reads	Clean Reads	Clean Bases	Q30 (%)	GC Content (%)	Total Mapped	Multiple Mapped	Uniquely Mapped
TS1	58,115,532	56,733,206	8.51 G	92.83	51.51	33,607,337 (59.24%)	404,573 (0.71%)	33,202,764 (58.52%)
TS2	45,612,568	44,369,036	6.66 G	92.36	50.87	26,015,839 (58.64%)	254,199 (0.57%)	25,761,640 (58.06%)
TS3	45,909,018	44,789,712	6.72 G	92.52	51.24	26,324,274 (58.77%)	294,976 (0.66%)	26,029,298 (58.11%)
TW1	51,833,172	49,968,538	7.5 G	92.88	51.62	28,571,508 (57.18%)	358,826 (0.72%)	28,212,682 (56.46%)
TW2	52,245,172	50,378,690	7.56 G	94.04	51.58	29,804,743 (59.16%)	324,963 (0.65%)	29,479,780 (58.52%)
TW3	59,370,820	57,202,160	8.58 G	94.1	52.02	32,617,622 (57.02%)	383,061 (0.67%)	32,234,561 (56.35%)

Note. TS indicates the leg muscles of the Shiotou goose, and TW means the leg muscles of the Wuzong goose (the same below).

**Table 5 genes-14-01048-t005:** The enriched KEGG pathways of DEGs in TS–VS–TW.

No.	Pathway	*p* Value	Genes (↑, ↓)
1	Metabolism of xenobiotics by cytochrome P450	0.017	↓*LOC106044188*, ↓*LOC106030438*
2	Glutathione metabolism	0.020	↓*LOC106030438*, ↓*ODC1*
3	Calcium signaling pathway	0.024	↑*CACNA1*, ↓*MYLK4*, ↑*SLC25A5*, ↑*ADORA2*
4	Vascular smooth muscle contraction	0.029	↑*CACNA1*, ↓*MYLK4*, ↑*ADORA2*

## Data Availability

The data sets supporting the results of this article were included within the article and its Appendix A.

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
