# Peer review of "Comparative Transcriptome Analyses of Leg Muscle during Early Growth between Geese (Anser cygnoides) Breeds Differing in Body Size Characteristics"

_genes, 2023, doi:10.3390/genes14051048_

Round 1

Reviewer 1 Report

General comments

This manuscript reports the growth phenotypes of two indigenous geese breeds, i.e., Shitou goouse and Wuzong goose, respectively, which show the difference in growth by estimating growth curve and comparative transcriptomic analyses of leg muscle to find out the underlying mechanisms.  

1. How many male and female were measured for growth trait. It is not clear in the Materials and Methods. Usually, males grow faster than females. Is there any way to include sex effect in the growth modeling?  

2.  Shitou and Wuzong males were used for RNA-sequencing.

Can these cause a bias to identify DEGs?  

Specific comments

1. In RNA-seq data, mapping rate to the reference genome is 56.35~58.52%.  It was also observed in the previous paper by the authors. Please include the reason of low mapping rate in discussion.

2. In Table 6, the correlation between RNA-seq and QRT-PCR validation data can be calculated. Please provide a correlation coefficient of this analysis.

Author Response

Thank you very much for recognizing the insights of our studies and providing further suggestions to improve our manuscript.

Reviewer 2 Report

The investigators compared growth rate and transcriptome of leg muscles between two breeds of geese. Overall quality of the manuscript meets the standard of MDPI. However, I've found some minor points that may help the investigators strengthen the manuscript. 

1) Please consider adding rationale on why you focused on leg muscles. Also, which muscle (name of the muscle) was collected.

2) line 84; Please add some information on animal handling. for example whether the geese were individually caged or free-range, etc.

3) line 124: please define fold change. If a gene showed decreased expression in Shitou relative to Wuzong, would the fold change show negative or positive numbers. If negative, how would the number define as differentially expressed?

4) Please recheck the quality of your tables and figures. Some of tables are difficult to read. Most of letters in all figures are difficult to read.

5) Please show all gene id in italics.

6) Based on the difference in growth rate between the two breeds which is very apparent, I am quite surprised not seeing any metabolic pathways or glucose/carbohydrate utilization. Do you have any speculation on this aspect?

7) Could you please add some information/speculation on why glutathione metabolism is among the enriched pathways?

8) I am wondering why you have focused on muscle regeneration in your discussion. Do you anticipate any muscle damage during the growth, particularly in Shitou? Based on my understanding, Ca2+ signaling is important with muscle metabolism, not just regeneration. 

Author Response

Thank you very much for recognizing the insights of our studies and providing further suggestions to improve our manuscript. We have listed the point-by-point replies below and incorporated our replies in the new edition of the manuscript. In the resubmitted manuscript, all the revisions have been highlighted in track and in red. Thanks again for your helpful suggestions.

Reviewer 3 Report

Title:  The title was appropriate  but in L#3,  add the scientific name of geese and correct the spelling of institutes in L#8.

Abstract: In L#13, remove the extra space among the Shitou, goose, and, & the

In L#24, remove the extra space among differentially, expressed genes.

In L#26, Insert a comma after MYLK4.

In L#30, use functions instead of function

Introduction: introduction summarizes relevant information.

Insert article  before growth (L#41).

Remove the extra space between of & goose in L#46.

Remove the extra spaces from multiple to environment (L#50).

In L#62, use the singular verb after muscle.

In L#65, use geese after Wuzong instead of goose.

In L#66, Insert a comma after Bertalanffy.

Materials and Methods: The methods are adequate. Check the grammar and remove the extra spaces in all sections (whole manuscript). In L#98, use the word logistics instead of logistic

In L#99, Insert the article before growth and goose.

Capitalized the T in the table (L#100)

In L#117, use a singular verb.

In L #118, remove the comma after nucleotides.

In L#124, capitalize the first word on ontology.

In L#127, Insert article before gene.

Result: Result in Section-Alignment with the Objectives, Materials & Methods, appropriate presentation, etc.)  The number of figures and tables was impressive. In L#145, capitalize the first word on the goose. Remove the spaces. In L#146, add the word before the body. Capitalize the first word on the goose (L#170).

In L#176, use the word grows instead of grow. In L#181, Insert the article  before Wuzong. In L#182, remove the comma after goose and extra space between shitou and goose. In L#183, add a comma after weeks. In L#190, Insert the article  before Shitou.

In L#193, add a hyphen between the fastest growing and use a singular verb.

In L#201, remove the comma after 92.36% and the preposition.

In L#211, remove extra space and use plural nouns (maps instead of the map).

In L#217, Insert a hyphen between downregulated. In L#219, use theirs instead of its.

In L#221, remove extra space between the Y-axis. In L#222, add space between the x-axis and displays. In L#223, remove the article before the transcripts. In L#224, remove the extra spaces. 

Discussion: 

Discussion-(Intellectual vigor, logical interpretation of results, and alignment of conclusion with the results, etc.). In L#281, insert a hyphen between week and old.

In L#282, Remove extra space, and insert a hyphen. In L#290: The phrase is wordy, remove so as.

In L# 305: The phrase is wordy, remove it in order.

In L#329: Remove the preposition.

In L#334: insert the article before migration.

In L#370: Change the punctuation and possessive pronoun.

In L#378: Insert Comma after triglyceride.

In L#410: insert the article before the sample.

Please avoid to used direct abbreviations, discuss it before.

Author Response

(The authors gave the same response as above.)

Reviewer 4 Report

It is hard to determine what is the real goal of the manuscript: comparison of the growth rate models or comparison of muscle transcriptome of two breeds of geese. The authors should be clear what is the aim of the manuscript. The title suggested that the transcriptome is the main aim of the paper while the abstract is pointing out towards growth models.

Additionally, why n=3 was selected for RNAseq?

Material and Methods: description of the bioinformatic analysis of the data should be more detailed. Provide all details of the analysis and all sources of the software used.

Table 3 and Figure 1 are redundant. Please select one.

Please indicate statistical differences in growth parameter between breeds.

Normal editing required.

Author Response

(The authors gave the same response as above.)

Round 2

Reviewer 2 Report

The manuscript looks significantly improved and can be accepted in its current form. 

Just my personal comments to the investigators about glutathione pathway. The results actually remind me of myopathies (WS, WB) in high-performance modern chickens. Perhaps, your results indicated different oxidative status within the leg muscles between those goose breeds. Anyway, I do not know for sure.

Reviewer 4 Report

I accept the responses from the authors

N/A